# Using Interpretation Methods for Model Enhancement

**Zhuo Chen, Chengyue Jiang, Kewei Tu**[*]
School of Information Science and Technology, ShanghaiTech University
Shanghai Engineering Research Center of Intelligent Vision and Imaging
{chenzhuo,jiangchy,tukw}@shanghaitech.edu.cn

## Abstract

In the age of neural natural language processing, there are plenty of works trying to derive interpretations of neural models. Intuitively, when gold rationales exist during training, one can additionally train the model to match its interpretation with the rationales. However, this intuitive idea has not been fully explored. In this paper, we propose a framework of utilizing interpretation methods and gold rationales to enhance models. Our framework is very general in the sense that it can incorporate various interpretation methods. Previously proposed gradient-based methods can be shown as an instance of our framework. We also propose two novel instances utilizing two other types of interpretation methods, erasure/replace-based and extractor-based methods, for model enhancement. We conduct comprehensive experiments on a variety of tasks. Experimental results show that our framework is effective especially in low-resource settings in enhancing models with various interpretation methods, and our two newly-proposed methods outperform gradient-based methods in most settings. Code is available at https://github.com/Chord-Chen-30/UIMER.

## 1 Introduction

Deep neural networks have been extensively used to solve Natural Language Processing (NLP) tasks and reach state-of-the-art performance. Due to the black-box nature of neural models, there are a lot of studies on how to interpret model decisions by giving **attribution scores** to input tokens, i.e., how much tokens in an input contribute to the final prediction. We can roughly group these interpretation methods into four categories, namely gradient-based (Ross et al., 2017; Smilkov et al., 2017), attention-based (Vashishth et al., 2019; Serrano and Smith, 2019), erasure/replace-based (Prabhakaran et al., 2019; Kim et al., 2020) and extractor-based (De Cao et al., 2020; Chan et al., 2022) methods.

*Corresponding author.

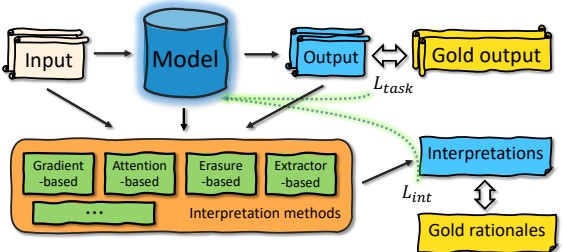

Figure 1: Our framework illustration utilizing interpretation methods to enhance models. The dotted green line indicates how the parameters of the model are optimized.

| Tokens | **add** | step | to | me | to | the | 50 | clásicos | **playlist** |
|---|---|---|---|---|---|---|---|---|---|
| Attribution scores | 0.7 | 0.3 | 0.1 | 0.1 | 0.1 | 0.3 | 0.0 | 0.1 | 0.6 |
| Gold rationales | 1 | 0 | 0 | 0 | 0 | 0 | 0 | 0 | 1 |

Figure 2: An example of *attribution scores* and *gold rationales*.

In some scenarios, we have access to gold rationales (input tokens critical for predicting correct outputs) during training, or have simple and fast approaches to obtaining gold rationales. In that case, it is intuitively appealing to additionally train the model such that its most attributed tokens match gold rationales (see an example in Fig. 2). In other words, when equipped with interpretation methods, we can train the model on where to look at in the input, in addition to the standard output-matching objective. This can be seen as injecting external knowledge embodied in rationales into the model and is especially beneficial to low-resource scenarios with few training data. This intuitive idea, however, has not been fully explored. There are only a few previous studies based on the gradient-based category of interpretation methods (Huang et al., 2021; Ghaeini et al., 2019; Liu and Avci, 2019) and they neither compare the utilization of different interpretation methods on model enhancement nor experiment on a comprehensive range of

tasks.

In this paper, we first propose a framework named UIMER that Utilizes Interpretation Methods and gold Rationales to improve model performance, as illustrated in Fig. 1. Specifically, in addition to a task-specific loss, we add a new loss that aligns interpretations derived from interpretation methods with gold rationales. We also discuss how the optimization of the task-specific loss and the new loss should be coordinated. For previous methods utilizing gradient-based interpretation for model enhancement (Huang et al., 2021; Ghaeini et al., 2019), we show that they can be seen as instances of our framework.

We then propose two novel instances of our framework based on erasure/replace-based and extractor-based interpretation methods respectively. Specifically, in the first instance, we utilize Input Marginalization (Kim et al., 2020) as the interpretation method in our framework, which computes attribution scores by replacing tokens with a variety of strategies and measuring the impact on outputs, and we design a contrastive loss over the computed attribution scores of rationales and non-rationales. In the second instance, we utilize Rationale Extractor (De Cao et al., 2020) as the interpretation method in our framework, which is a neural model that is independent of the task model and trained with its own loss. We again design a contrastive loss over the attribution scores computed by the extractor and in addition, design a new training process that alternately optimizes the task model (using the task-specific loss and our contrastive loss) and the extractor (using its own loss).

In summary, our main contributions can be summarized as follows: (1) We propose a framework that can utilize various interpretation methods to enhance models. (2) We utilize two novel types of interpretation methods to enhance the model under our framework. (3) We comprehensively evaluate our framework on diversified tasks including classification, slot filling and natural language inference. Experiments show that our framework is effective in enhancing models with various interpretation methods, especially in the low-resource setting.

## 2 Related Work

### 2.1 Interpretation Methods

Interpretation methods aim to decipher the black-box of deep neural networks and have been well-studied recently. Our framework aims to utilize these various interpretation methods for model enhancement. In these interpretation methods, attribution scores are calculated to indicate the importance of input tokens. According to different calculations of attribution scores, we can generally group interpretation methods into the following four categories.

**Gradient-based methods** Gradient-based interpretation methods are quite popular and intuitive. Li et al. (2016) calculates the absolute value of the gradient w.r.t each input token and interprets it as the sensitiveness of the final decision to the input. Following that, Ghaeini et al. (2019) extends the calculation of gradient to intermediate layers of deep models. Sundararajan et al. (2017) proposes a better method that calculates *Integrated Gradients* as explanations of inputs.

**Attention-based methods** The attention mechanism calculates a distribution over input tokens, and some previous works use the attention weights as interpretations derived from the model (Wang et al., 2016; Ghaeini et al., 2018). However, there is no consensus as to whether attention is interpretable. Jain and Wallace (2019) alters attention weights and finds no significant impact on predictions, while Vig and Belinkov (2019) finds that attention aligns most strongly with dependency relations in the middle layers of GPT-2 and thus is interpretable.

**Erasure/replace-based methods** The general idea is quite straightforward: erase or replace some words in a sentence and see how the model's prediction changes. Li et al. (2016) proposes a method to analyze and interpret decisions from a neural model by observing the effects on the model of erasing various parts of the representation, such as input word-vector dimensions, intermediate hidden units, or input words. Kim et al. (2020) gives a new interpretation method without suffering from the out-of-distribution (OOD) problem by replacing words in inputs and it reaches better interpretability compared with traditional erasure-based methods.

**Extractor-based methods** An extractor-based method typically uses an extra model to extract words that the task model pays attention to. De Cao et al. (2020) introduces "Differentiable Masking" which learns to mask out subsets of the inputs while maintaining differentiability. This decision is made with a simple model based on intermediate hid-

den layers and word embedding layers of a trained model. Chen and Ji (2020) proposes the variational word mask method to learn to restrict the information of globally irrelevant or noisy word level features flowing to subsequent network layers.

## 2.2 Utilizing Interpretation Methods to Enhance Models

Some previous work studies whether interpretation methods can be utilized to enhance models. Du et al. (2019) is closely related to one of our contributions (i.e., UIMER-IM in Sec. 3.2). However, their model suffers from the OOD problem and does not consistently outperform baselines in their experiments. Ghaeini et al. (2019) designs a simple extra objective that encourages the gradient of input to have a positive effect on ground truth. They find it useful in event-extraction and Cloze-Style question-answering tasks. To handle *Non-Important Rationales* and exploit *Potential Important Non-rationales*, Huang et al. (2021) designs their method following the idea that rationales should get more focus than non-rationales and only part of the rationales should attain higher focus. They focus on text classification tasks and find that, despite the rationale annotations being insufficient and indiscriminative, their method can bring improvements. However, these previous studies neither compare the utilization of various interpretation methods on model enhancement nor conduct experiments on a comprehensive range of tasks.

## 3 Method

We propose a framework UIMER that utilizes an interpretation method to enhance models based on gold rationales on training data.

**Setup**  Consider a training example with input $x$ and gold output $y$. We also have access to gold rationales $g$ of input $x$ indicating the subset of tokens that are critical for predicting the correct output $y$. The gold rationales $g$ can be annotated by humans or generated automatically from external knowledge sources.

In our setup, $g$ is encoded as a $0/1$ vector with $g_i = 1$ indicating that $x_i$ is a gold rationale and $g_i = 0$ otherwise. Our method, however, can be easily extended to handle $g$ encoded with real numbers indicating the importance of a token.

Given a model tasked with predicting output $y$ from input $x$, an interpretation method produces attribution scores $a$ for input $x$. $a$ can be defined on different levels of granularity. In common cases, $a_i$ and $x_i$ are one-to-one and $a_i$ is defined by a measure calculated by the interpretation method based on the model indicating the importance of token $x_i$ for the model in producing its output. For example, in gradient-based interpretation methods, $a_i$ is usually some function of the gradient of $x_i$ in the model. A higher magnitude of the gradient implies higher importance of input $x_i$.

**Learning Objective**  Apart from the original task-specific learning objective $L_{task}$, our framework introduces an extra learning objective $L_{int}$ that embodies the idea of aligning attribution scores $a$ with gold rationales $g$. The overall objective on one example $x$ takes the form of:

$$L_\theta = L_{task}(\boldsymbol{x}, y) + \alpha L_{int}(\boldsymbol{a}, \boldsymbol{g}) \qquad (1)$$

where $\theta$ is the model parameter and $\alpha$ is a coefficient balancing the two objectives.

**Warm-up Training**  $L_{int}$ can be seen as measuring whether the model pays attention to the gold rationale words in the input. We deem that compared to teaching a randomly initialized model focusing more on the task-specific rationale words, teaching a model with task knowledge is more effective and reasonable because it might be better at recognizing task-specific rationales with the help of task knowledge rather than rote memorization. Thus, instead of optimizing $L_\theta$ at the very beginning, our framework requires the model to be well or at least halfway trained before training on the objective $L_{int}$, and during warm-up training, only the objective of the task is optimized. The empirical results (in Sec. 4.2) also support our intuition and show that warm-up training is effective.

In the following subsections, we introduce three instances of our framework. The first utilizes gradient-based interpretation methods and subsumes a few previous studies as special cases. The second and third are new methods proposed by us utilizing erasure/replace-based and extractor-based interpretation methods respectively.

## 3.1 Utilizing Gradient-Based Methods

As introduced in Sec. 2.2, some previous studies utilize gradient-based (GB) interpretation methods to enhance models. They can be seen as instances of our framework, hence denoted as UIMER-GB.

In this type of methods, attribution score $\boldsymbol{a}$ is usually defined by a function $f$ of the gradient of input $\boldsymbol{x}$:

$$\boldsymbol{a} = f\left(\frac{\partial J}{\partial \boldsymbol{x}}\right) \qquad (2)$$

where usually $J$ refers to the training objective or the task model's output.

In general, $L_{int}$ is defined as a constraint on the gradients of gold rationales:

$$L_{int}(\boldsymbol{a}, \boldsymbol{g}) = D(\boldsymbol{a}, \boldsymbol{g}) \qquad (3)$$

where $D$ is usually a distance function that calculates the error of how $\boldsymbol{a}$ approaches $\boldsymbol{g}$.

In Ghaeini et al. (2019)'s work, $f$ is a function that takes the sum of the gradients of each input embedding dimension and $D$ is to take the sum of the gradients of rationale words. In Huang et al. (2021)'s work, $f$ is the $L_1$ norm that sums up the absolute value of gradients over the input embedding dimensions and $D$ is designed in various ways to give rationale words higher attribution scores.

### 3.2 Utilizing Erasure/Replace-Based Methods

We incorporate an erasure/replaced-based interpretation method, "Input Marginalization" (IM) (Kim et al., 2020), into our framework in this subsection and name this instance UIMER-IM. We first define the attribution score produced by IM and then define and introduce how to calculate $L_{int}$. Other erasure/replace-based methods can be integrated into our framework in a similar way.

#### 3.2.1 Attribution Score by Input Marginalization

Define $p_\theta(y|\boldsymbol{x})$ as the probability of the gold output that the model predicts. To calculate the attribution score of token $x_i$, a new set of sentences $\boldsymbol{S}$ needs to be generated, with the size of $\boldsymbol{S}$ being a hyperparameter. Denote $\boldsymbol{x}'_u$ as a new sentence with $x_i$ replaced by some other token $u$. Denote $q(\boldsymbol{x}'_u)$ as the probability of replacing $x_i$ with $u$ to obtain $\boldsymbol{x}'_u$, which can be determined by different strategies[1]:

1. BERT: Replace $x_i$ by [MASK] to obtain the masked language model probability of $u$.

2. Prior: $q(\boldsymbol{x}'_u) = \text{count}(u)/N$, where $\text{count}(u)$ is the number of times token $u$ appears in corpus and $N$ is the corpus size.

3. Uniform: $q(\boldsymbol{x}'_u) = \frac{1}{|V|}$ where $|V|$ is the vocabulary size.

We follow one of the strategies to sample set $\boldsymbol{S}$ based on $q(\boldsymbol{x}'_u)$ [2], and define $a_i$ as:

$$a_i = \log_2(\text{odds}(p_\theta(y|\boldsymbol{x}))) - \log_2(\text{odds}(m)) \quad (4)$$

where

$$m = \sum_{\boldsymbol{x}'_u \in \boldsymbol{S}} q(\boldsymbol{x}'_u) p_\theta(y|\boldsymbol{x}'_u)$$

$$\text{odds}(p) = p/(1-p)$$

For inputs with only one gold rationale word, computing and optimizing the attribution score is easy. However, there might be more than one rationale in general and the calculation of the attribution score of each rationale token becomes impractical when the input length and the number of rationales get larger. Therefore, we extend the token attribution score defined in the original IM method to multi-token attribution score which can be more efficiently computed.

Formally, for input $\boldsymbol{x}$ with more than one rationale word, denote $\boldsymbol{x}'_R$ as a new sentence in which all rationale words are replaced, and $\boldsymbol{x}'_N$ as a new sentence in which the same number of non-rationale words are replaced.[3] The way to generate one $\boldsymbol{x}'_R$ is by replacing one rationale word at a time using the strategies mentioned before. We denote the score of replacing $\boldsymbol{x}$ to $\boldsymbol{x}'_R$ as $q(\boldsymbol{x}'_R)$, and $q(\boldsymbol{x}'_R)$ is calculated as the average of the replacing probabilities of rationale words using a certain replacing strategy[4]. Similarly, $\boldsymbol{x}'_N$ is generated and $q(\boldsymbol{x}'_N)$ can be defined. We repeat this generating process and denote the set of generated $\boldsymbol{x}'_R(\boldsymbol{x}'_N)$ as $\boldsymbol{S^R}(\boldsymbol{S^N})$ with the size of $\boldsymbol{S^R}(\boldsymbol{S^N})$ being a hyperparameter. Then the attribution scores $a_R$ for the entire set of rationales and $a_N$ for the same number

---

[1]In our experiment, we also include a strategy "MASK", which means $x_i$ is simply replaced by the [MASK] token and $q(\boldsymbol{x}'_{[MASK]}) = 1$

[2]When $q(\boldsymbol{x}'_u)$ is determined by BERT strategy, we follow the method IM to construct $\boldsymbol{S}$. Words with the highest probabilities are selected rather than sampled to replace $x_i$.

[3]The same number of non-rationales to be replaced are chosen randomly from the input.

[4]Here we deviate from the calculation of Kim et al. (2020) that multiplies the probabilities. If there are many rationale words in a sentence and we take the product of probabilities, the sentence with the most probable words replaced by BERT will have a dominant probability compared with others, which often degenerates the calculation of attribution score.

of non-rationales are defined as:

$$a_R = \log_2(\text{odds}(p_\theta(y|\boldsymbol{x}))) - \log_2(\text{odds}(m_R))$$

$$a_N = \log_2(\text{odds}(p_\theta(y|\boldsymbol{x}))) - \log_2(\text{odds}(m_N))$$

$$m_R = \sum_{\boldsymbol{x}'_R \in \boldsymbol{S^R}} q(\boldsymbol{x}'_R) p_\theta(y|\boldsymbol{x}'_R)$$

$$m_N = \sum_{\boldsymbol{x}'_N \in \boldsymbol{S^N}} q(\boldsymbol{x}'_N) p_\theta(y|\boldsymbol{x}'_N)$$

### 3.2.2 Definition of $L_{int}$ in UIMER-IM

For a given input $\boldsymbol{x}$ and gold rationale $\boldsymbol{g}$, with $a_R$ and $a_N$ defined, we expect the attribution score of rationale words to be higher than that of non-rationale words. Thus, we design a contrastive margin loss:

$$L_{int}(\boldsymbol{a}, \boldsymbol{g}) = max(a_N - a_R + \epsilon, 0) \quad (5)$$

where $\epsilon$ is a positive hyperparameter controlling the margin. Here $\boldsymbol{a}$ is not defined w.r.t each token, and it refers to the attribution score for multi-tokens. Note that when calculating $a_N - a_R$, the term $\log_2(\text{odds}(p_\theta(y|\boldsymbol{x})))$ is canceled out and does not need to be computed. We choose to use the margin loss instead of simply maximizing $a_R$ and minimizing $a_N$ because in many cases non-rationale words may still provide useful information and we do not want to eliminate their influence on the model.

### 3.3 Utilizing Extractor-Based Methods

In this section, we incorporate "DiffMask" (DM) (De Cao et al., 2020), an extractor-based interpretation method into our framework and name this instance UIMER-DM. Other extractor-based interpretation methods can be integrated into UIMER in a similar way.

### 3.3.1 Attribution Score by the Extractor

In DM, the attribution scores $\boldsymbol{a}$ are produced by a simple extractor model $Ext_\phi$ parameterized by $\phi$.

$$\boldsymbol{a} = Ext_\phi(Enc(\boldsymbol{x})) \quad (6)$$

where $Enc(\boldsymbol{x})$ refers to the encoding of input $\boldsymbol{x}$ produced by an encoder model, and $\boldsymbol{a}$ is composed of real numbers in the range of 0 to 1. Here $\boldsymbol{a}$ is defined one-to-one w.r.t. each token in $\boldsymbol{x}$. The extractor needs to be trained by its own objective $L_\phi^{\text{DM}}$ composed of 2 parts:

1. A term to encourage masking the input (or hidden states) as much as possible.

2. A term to constrain the changes of task model's prediction after feeding the masked input (or hidden states).

### 3.3.2 Definition of $L_{int}$ in UIMER-DM

With attribution scores defined in DM, we define a contrastive loss $L_{int}$ as follows:

$$L_{int}(\boldsymbol{a}, \boldsymbol{g}) = \sum_{i:g_i=1} \left( \min \left( \frac{a_i}{\max\limits_{j:g_j=0} a_j} - 1, 0 \right) \right)^2 \quad (7)$$

Intuitively, we encourage the attribution score of rationale words to be higher than the attribution score of non-rationale words. The loss will be zero as long as the maximum attribution score of non-rationale words is lower than the attribution score of any rationale word.

### 3.3.3 Training in UIMER-DM

When training UIMER-DM, there are two objectives and two sets of parameters, $L_\theta$ (Eq. 1) for model parameter $\theta$ and $L_\phi^{\text{DM}}$ for extractor parameter $\phi$. Intuitively, the two sets of parameters should not be optimized simultaneously. That is because our framework requires an accurate interpretation model (i.e., the extractor here). If the extractor is trained at the same time with the model, then since the model keeps changing, there is no guarantee that the extractor could keep pace with the model, and hence its interpretation of the model may not match the latest model, breaking the requirement of our framework.

We adopt the following training schedule to circumvent the problem. First, we follow the warm-up strategy and train the model. After that, we alternate between two steps: (1) optimizing the extractor parameters $\phi$ w.r.t. $L_\phi^{\text{DM}}$ with the model parameters $\theta$ frozen; (2) optimizing the model parameters $\theta$ w.r.t. $L_\theta$ with the extractor parameters $\phi$ frozen. The number of rounds that we alternately optimize $\phi$ and $\theta$ and the number of epochs in each round are hyperparameters.

## 4 Experiment

### 4.1 Experimental Settings

**Datasets** To evaluate our framework, we experiment with all the methods introduced in the previous section on three tasks: Intent Classification (IC), Slot Filling (SF) and Natural Language Inference (NLI). The three tasks take the forms of single sentence classification, sequence labeling

| Task | Dataset | Source of Rationales | Rationales |
|------|---------|---------------------|------------|
| IC | SNIPS | Human annotation | **add** step to me to the 50 clásicos **playlist** |
| SF | SNIPS | Regular Expression | **rate** the current essay 2 **out of** 6 |
| NLI | e-SNLI | Given in the dataset | Premise: children **smiling** and waving at camera 
 Hypothesis: the kids are **frowning** |

Table 1: Examples of rationales on the three tasks in our experiments.

and sentence pair classification respectively. For Intent Classification and Slot Filling, we adopt the SNIPS (Coucke et al., 2018) dataset. For Natural Language Inference, we adopt the e-SNLI (Camburu et al., 2018) dataset. **SNPIS** is widely used in NLU research (Jiang et al., 2021a; Chen et al., 2019). It is collected from SNIPS personal voice assistant. There are 13084, 700 and 700 samples in the training, development and test sets respectively, and there are 72 slot labels and 7 intent types. **e-SNLI** is a large dataset extended from the Stanford Natural Language Inference Dataset (Bowman et al., 2015) to include human-annotated natural language explanations of the entailment relations. There are 549367, 9842, and 9824 samples in the training, development and test sets respectively.

**Rationales** We give examples of rationales and show how they are obtained in Table 1. For the Intent Classification task, we ask one annotator to construct a set of keywords for each intent type based on the training set. This only takes less than 15 minutes. For the Slot Filling task, we use 28 regular expressions with simple patterns which reference Jiang et al. (2021b)[5] to match the sentences in the SNIPS dataset and regard matched tokens as rationales. The job takes less than 30 minutes (less than 1 minute each on average). The complete rationale sets for Intent Classification and Slot Filling task are shown in App. 8.1. For e-SNLI, we use the explanations provided by Camburu et al. (2018).

**Baselines** For Intent Classification task, we choose *BERT-base-uncased*[6] + *Softmax* as our base model[7]. For Slot Filling, we choose *BERT-base-uncased + CRF* as our base model. For Natural Language Inference, we prepare the data into the form "*[CLS] premise [SEP] hypothesis [SEP]*", feed it into *BERT-base-uncased*, and apply a linear layer to the hidden state of the *[CLS]* token to score the entailment. These base models are natural base-

lines. We also regard the two previously-proposed gradient-based methods introduced in Sec. 3.1 as stronger baselines.

**Training Settings** We mainly focus on low-resource settings with limited training examples, which is when external resources such as rationales can be most beneficial. With less training data, there is an increasing return of utilizing rationales. For Slot Filling and Intent Classification, we compare our methods and baselines on 1-shot, 3-shot, 10-shot, 30-shot and 100% of training data. For Slot Filling, $n$-shot means that we sample training examples such that each slot label appears at least $n$ times in the sampled examples. For Natural Language Inference, we compare our framework and baselines with 100 and 500 training samples from e-SNLI. Hyperparameters are tuned on the development set and we report the average result of 4 random seeds on the test set. Detailed data about hyperparameters is shown in App. 8.3.

### 4.2 Main Results

We present the main results in all the settings in Table 2. First, from the Mean column, we see that gradient-based methods (UIMER-GB Ghaeini et al. (2019); Huang et al. (2021)) reach better performance than the base model, and all variants of our proposed UIMER-IM and UIMER-DM methods outperform both UIMER-GB and base models.

For UIMER-IM, its variants achieve the best performance in eight of the twelve settings and the second best in the rest of settings, suggesting its general applicability. The "+Uniform" variant of UIMER-IM can be seen to clearly outperform the other variants in the 1/3-shot settings on Intent Classification and we analyze the potential reason behind this in App. 8.2. UIMER-IM with the "BERT (warm.)" variant brings a 14.86% gain for the NLI 100-example setting.

For UIMER-DM, it achieves the best performance in the 1/3-shot settings and is competitive in the 10-shot setting on Slot Filling, which indicates

---

[5]These regular expressions are designed to extract slots.

[6]https://huggingface.co/bert-base-uncased

[7]We implement it based on *JointBERT*

| Model | | IC (Acc.) | | | | | SF (F1) | | | | | NLI (Acc.) | | Mean |
|---|---|---|---|---|---|---|---|---|---|---|---|---|---|---|
| | | 1 | 3 | 10 | 30 | full | 1 | 3 | 10 | 30 | full | 100 | 500 | - |
| BaseModel | | 65.71 | 79.18 | 91.00 | 93.79 | 97.64 | 38.14 | 50.97 | 67.05 | 81.70 | 95.22 | 54.03 | 62.84 | 73.11 |
| UIMER-GB Ghaeini et al. (2019) | | 65.71 | 79.14 | 91.82 | 94.18 | 97.92 | 37.77 | 51.69 | 67.57 | 82.16 | **95.99** | 67.13 | 69.77 | 75.07 |
| UIMER-GB Huang et al. (2021) | + Base | 67.04 | 83.04 | 91.43 | 94.57 | 98.21 | 39.02 | 50.66 | 67.11 | 80.20 | 95.25 | 66.15 | 68.57 | 75.10 |
| | + Gate | 67.14 | 82.01 | 91.39 | 94.07 | 98.07 | 37.84 | 51.63 | 67.34 | 80.89 | 95.50 | 66.06 | 68.31 | 75.02 |
| | + Order | 65.28 | 81.82 | 90.82 | 94.36 | 98.11 | 38.18 | 50.99 | 67.55 | 81.08 | 95.39 | 68.44 | 68.57 | 75.05 |
| | + (Gate+Order) | 67.71 | 80.25 | 92.11 | 94.39 | **98.39** | 38.86 | 51.73 | 67.76 | 80.55 | 95.77 | 65.10 | 65.84 | 74.87 |
| UIMER-IM | + MASK | 69.85 | 83.17 | 91.18 | 93.86 | 98.00 | 38.68 | 52.47 | 69.27 | 81.67 | 95.83 | 63.36 | 70.04 | 75.61 |
| | + BERT | 70.61 | 83.93 | 91.78 | **94.61** | 97.86 | 39.28 | 51.96 | 69.22 | 81.85 | 95.77 | 62.08 | 69.03 | 75.67 |
| | + Prior | 73.71 | 83.93 | 91.00 | 93.96 | 97.89 | 38.07 | 51.69 | 68.31 | 81.97 | 95.28 | 66.56 | 70.32 | 76.06 |
| | + Uniform | 73.32 | 86.04 | 91.64 | 94.00 | 98.04 | 39.31 | 51.37 | 69.02 | 81.69 | 95.49 | 64.34 | 69.19 | 76.12 |
| UIMER-IM | + MASK (warm.) | 70.82 | 82.96 | 92.43 | 94.04 | 98.25 | 38.60 | 51.55 | 67.93 | 82.25 | 94.89 | 68.79 | 69.95 | 76.04 |
| | + BERT (warm.) | 70.93 | 82.71 | 92.00 | 94.32 | 97.82 | 39.53 | 52.83 | 68.81 | 82.58 | 95.74 | **68.89** | 69.18 | 76.28 |
| | + Prior (warm.) | 73.82 | 83.11 | 91.93 | 94.07 | 97.82 | 38.24 | 51.68 | 68.26 | **82.66** | 95.82 | 68.25 | **71.82** | 76.46 |
| | + Uniform (warm) | **75.79** | **86.29** | 91.67 | 94.32 | 98.11 | 38.71 | 52.18 | 68.21 | 82.17 | 95.39 | 67.58 | 69.79 | **76.68** |
| UIMER-DM | One-pass | 66.75 | 84.42 | 91.53 | 93.78 | 97.96 | 39.86 | 52.87 | 67.28 | 81.90 | 95.22 | 63.03 | 66.91 | 75.13 |
| | Multi-round | 70.21 | 85.86 | 91.92 | 94.00 | 97.96 | **41.32** | **53.10** | 69.26 | 82.00 | 95.54 | 65.44 | 67.60 | 76.18 |

Table 2: Evaluation of our framework on three tasks. Underlines mark the results of our UIMER-IM/DM that outperform the base model and UIMER-GB methods. **Boldface** marks the best results among all the methods. The Mean column gives the average of each row.

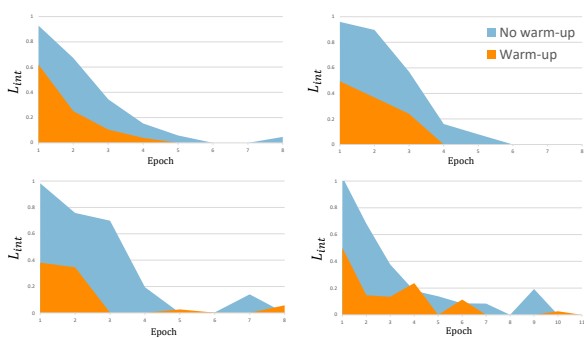

Figure 3: The curves of $L_{int}$ with and without warm-up training on 1-shot Intent Classification over 4 random seeds.

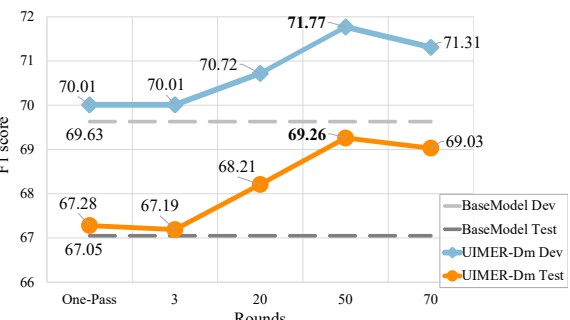

Figure 4: Ablation study of multi-round training on 10-shot Slot Filling.

that an extra extractor might be more capable of locating rationales than other interpretation methods for structured prediction problems. Applying multi-round training to UIMER-DM can be seen to clearly improve the performance in almost all the settings of all three tasks. We conduct an ablation study on the effect of multi-round training of UIMER-DM in Sec. 5.2.

From the results, it is also evident that in general, the performance gap between any instance method of our framework and the base model becomes larger with less data, implying the usefulness of rationales when training data cannot provide sufficient training signals for model enhancement.

## 5 Analysis

### 5.1 Warm-up Training

We conduct an ablation study on the effect of warm-up training of our method UIMER-IM in the 1-shot setting of the Intent Classification task. In particular, we inspect the value of objective $L_{int}$ during training with and without warm-up training, as shown in Fig. 3. It can be seen that with warm-up training, the $L_{int}$ objective starts with a lower value and converges to zero faster, verifying the benefit of warm-up training.

### 5.2 Multi-Round Training

In our method UIMER-DM, we propose an asynchronous training process that trains the model and the extractor asynchronously for multiple rounds. We conduct a study on the effectiveness of the multi-round training process on 10-shot Slot Filling and the results are shown in Fig. 4. We observe that by iteratively and alternately training the extractor and the model, performances on both the development and the test sets show an upward trend with more rounds. Note that we use fewer training epochs (around 10 for the extractor and 20 for the

|  | add karusellen to jazz brasileiro | Prediction |
|---|---|---|
| BaseModel +IM | $a_R : 1.787 \; < \; a_N : 2.355$ | ✘ PlayMusic |
| UIMER-IM | $a_R : \mathbf{6.098} \; > \; a_N : 0.544$ | ✔ AddToPlaylist |

|  | what are the movie schedules | Prediction |
|---|---|---|
| BaseModel +IM | $a_R : 1.808 \; < \; a_N : 2.172$ | ✘ SearchCreativeWork |
| UIMER-IM | $a_R : \mathbf{2.424} \; > \; a_N : 1.465$ | ✔ SearchScreeningEven |

|  | rate | the | current | chronic | five | stars | | | Prediction |
|---|---|---|---|---|---|---|---|---|---|
| BaseModel +DM | 0.99 | 0.99 | 0.99 | 0.72 | 0.99 | 0.95 | | | ✘ SearchCreativeWork |
| UIMER-DM | 0.00 | 0.00 | 0.00 | 0.00 | 0.00 | **0.99** | | | ✔ RateBook |

|  | play | the | newest | music | by | evil | jared | hasselhoff | Prediction |
|---|---|---|---|---|---|---|---|---|---|
| BaseModel +DM | 0.99 | 0.99 | 0.99 | 0.99 | 0.99 | 0.00 | 0.99 | 0.99 | ✔ PlayMusic |
| UIMER-DM | **0.99** | 0.00 | 0.00 | 0.00 | 0.00 | 0.00 | 0.00 | 0.00 | ✔ PlayMusic |

Table 3: A case study that analyzes the task performance and quality of interpretations of the base model and our methods UIMER-IM&DM. "BaseModel +IM/DM Attr.": The attribution scores produced by base model and IM/DM interpretation method. UIMER-IM/DM: The attribution scores produced by our framework.

model) in each round of multi-round training than one-pass training. 3-round training does not outperform one-pass training simply because it has few total training epochs.

### 5.3 Case Study

To show that our framework is able to both enhance the model in task performance and give better interpretations, we conduct a case study on 1-shot Intent Classification setting. From the first two examples in Table 3, we can see that the base model can neither predict the correct intent label of the sentence nor produce good interpretations (the attribution scores of non-rationales are higher than the scores of rationales); in comparison, our UIMER-IM fixes both problems. In the last two examples, we show that UIMER-DM succeeds in lowering the attribution scores of irrelevant parts in the input and producing high scores for some or all of the rationales. It can also be seen that the extractor trained on the base model in 1-shot settings views most of the inputs as being important, while the extractor in UIMER-DM is much more parsimonious and precise.

### 5.4 Relation Between Attribution Score and Performance

In this section, we study how the model performs on the test set when it succeeds or fails to give rationale words higher attribution scores. We conduct experiments on 1-shot Intent Classification and calculate the accuracy while giving rationale words higher or lower attribution scores than non-

|  | Acc. $a_R > a_N$ | Acc. $a_R \le a_N$ | % $a_R > a_N$ | Acc. |
|---|---|---|---|---|
| BaseModel +IM | 68.86 | 37.86 | 81.57 | 65.71 |
| UIMER-IM | **77.99** | 36.59 | 85.33 | 75.79 |
| BaseModel +DM | 72.44 | 52.44 | 58.57 | 65.71 |
| UIMER-DM | **75.75** | 16.52 | 81.28 | 70.21 |

Table 4: Performances when the model gives higher attribution scores to rationale words or not.

rationales, as shown in the first and second columns in Table 4. For methods with the DM interpretation method, $a_R$ is calculated by averaging the attribution scores for all rationale words in $\boldsymbol{x}$ and $a_N$ is calculated by averaging the attribution scores for all non-rationale words. We can see that when our UIMER-IM/DM method correctly recognizes rationale words, it reaches higher accuracy than the base model, which suggests that helping models pay more attention to rationales can additionally improve the task performance.

### 6 Conclusion

Though many interpretation methods are studied for deep neural models, only sporadic works utilize them to enhance models. In this paper, we propose a framework that can utilize various interpretation methods to enhance models. We also propose two novel instances utilizing two other types of interpretation methods for model enhancement. In addition, we discuss how the optimization of the task-specific loss and the new loss should

be coordinated. Comprehensive experiments are conducted on a variety of tasks including Intent Classification, Slot Filling and Natural Language Inference. Experiments show that our framework is effective in enhancing models with various interpretation methods especially in the low-resource setting, and our two newly-proposed methods outperform gradient-based methods in most settings. For future work, we plan to extend our framework to utilize more forms of rationales and additional interpretation methods.

# 7 Limitations

It can be inferred from the result that the two newly introduced methods do not give the best performance in rich-resource settings. We prospect that method UIMER-IM plays a role in incorporating the information of rationales by introducing more similar inputs to the model when the training data is scarce. However, when training data is sufficient enough for the task, the effect of this way to supply information on rationales is reduced. For method UIMER-DM, it also does not perform the best in rich-resource settings. We attribute the ordinary performance of UIMER-DM to that with rich data, most knowledge can be implicitly learned by the model, and injecting gold rationale doesn't help.

## Acknowledgement

This work was supported by the National Natural Science Foundation of China (61976139).

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

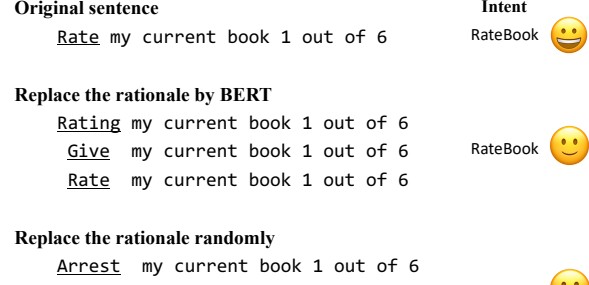

Figure 5: The underline marks a rationale. Replace the rationale by BERT v.s. randomly.

Yequan Wang, Minlie Huang, Xiaoyan Zhu, and Li Zhao. 2016. Attention-based lstm for aspect-level sentiment classification. In *Proceedings of the 2016 conference on empirical methods in natural language processing*, pages 606–615.

## 8 Appendix

### 8.1 Patterns to match rationales

In our experiments, we adopt simple patterns to match rationales. For Intent Classification, a dictionary is constructed, as shown in Table 5 (upper), mapping intent labels to rationales, and for Slot Filling, a group of Regular Expressions, as shown in Table 5 (below), is used to extract rationales. Both tasks are derived from SNIPS dataset that contains 13084 examples, and obtaining both the dictionary and the Regular Expressions is simple and fast. Please note that though the table below seems a little complex, the gray parts are just the syntax of Regular Expressions. Only the black parts contain rational information.

### 8.2 Why is "+Uniform" Good

In experimenting with Sec. 3.2, we found that replacing rationales/non-rationales randomly, i.e. the "Uniform" strategy often produces a better result even than the "BERT" strategy, despite the fact that compared to replacing tokens randomly, a pretrained BERT can apparently produce more fluent sentences. Here we give an example in Intent Classification task that possibly explains the cause of this fact in Fig. 5.

As stated in Sec. 3.2, we aim to minimize $L_{int}$ defined in Eq. 5 in which we want to lower the latter term in $a_R$, and it is a weighted sum of the predictions of these sentences generated by BERT on the ground truth intent. However, we see that BERT pretty much keeps the original meaning of

the sentence in this example. Thus lowering the latter term in $a_R$ seems to be a contradiction to the origin task. In contrast, if we look at the sentences generated by replacing the rationale randomly they contain much less information about the ground truth intent, and minimizing $L_{int}$ seems to be more reasonable.

### 8.3 Hyperparameters

For Intent Classification and Slot filling task, the learning rate $L_\theta$ is tuned among {8e-5, 1e-4, 2e-4, 4e-4} in 1/3/10/30-shot settings and {1e-5, 2e-5} in full training data setting. For Natural Language Inference task, the learning rate of $L_\theta$ is tuned among {8e-5, 1e-4, 3e-4}. In all three tasks, $\alpha$ is tuned in the range [0.001, 20]. $\epsilon$ is tuned in the range [0.01, 10]. We use AdamW optimizer (Loshchilov and Hutter, 2018) and "linear schedule with warmup" scheduler. Detailed hyperparameters are shown in Table 6-8.

### 8.4 Result with std.

We show the result with unbiased estimation of standard deviation in Table 9 in all few-shot settings.

| Task | Intent Label (keys) | **Rationales (values)** |
|---|---|---|
| | AddToPlaylist | [add, playlist, album, list] |
| | BookRestaurant | [book, restaurant, reservation, reservations] |
| | GetWeather | [weather, forecast, warm, freezing, hot, cold] |
| IC | PlayMusic | [play, music, song, hear] |
| | RateBook | [rate, give, star, stars, points, rating, book] |
| | SearchCreativeWork | [find, show, called] |
| | SearchScreeningEvent | [movie, movies, find, theatres, cinema, cinemas, film, films, show] |

| Task | **Regular Expressions** |
|---|---|
| | .*(?P<rationale>find\|looking for\|show\|download\|get) (?P<rationale1>me) (?P<rationale2>a\|the).*(called).* |
| | .*(?P<none>[0-5]\|zero\|one\|two\|three\|four\|five) (?P<rationale>points\|stars).* |
| | .*(?P<rationale>a rating of) (?P<none>[0-5]\|zero\|one\|two\|three\|four\|five).* |
| | .*(?P<rationale>rate\|give) .*(?P<rationale1>out of).* |
| | .*(?P<none>[0-5]\|zero\|one\|two\|three\|four\|five) (?P<rationale>out of) (?P<none1>6\|six).* |
| | .*(this\|current)? (?P<rationale>book\|novel\|movie schedule\|movie schedules\|album\|movie schedules\| movie times\|essay\|textbook\|tv show\|saga\|trailer\|photograph\|picture\|television show\|game\|painting\| tv series\|soundtrack\|song\|movie\|saga\|series\|chronicle).* |
| | .*(?P<rationale>add) .*(?P<rationale1>to).* |
| | .*(?P<rationale>add\|put) .*(?P<rationale1>to my) (?P<rationale2>playlist)?.* |
| | .*(?P<rationale>play playlist).* |
| | .*(?P<rationale>my) .*(?P<rationale1>playlist).* |
| SF | .*(?P<rationale>song\|album\|track\|tune\|artist\|soundtrack) (?P<rationale1>by)?.* |
| | .*(?P<rationale>weather\|sunny\|forecasted\|forecast) .*(?P<rationale1>in).* |
| | .*(?P<rationale>what is the weather).* |
| | .*(?P<rationale>weather\|weather forecast).* |
| | .*(?P<rationale>book) (?P<none>a).* |
| | .*(?P<rationale>restaurant\|bar\|brasserie\|pub\|taverna\|food truck\|cafeteria).* |
| | .*(?P<rationale>nearest\|closest\|nearby\|close by\|in the neighborhood\|in the area).* |
| | .*(?P<rationale>table\|seats\|reservation\|restaurant\|spot) .*(?P<rationale1>for) .*(?P<rationale2>people)?.* |
| | .*(?P<rationale>movie house\|cinema\|movie theatre).* |
| | .*(?P<rationale>when is\|what time is\|find me\|where is\|is\|see\|watch).* (?P<rationale1>playing\|showing).* |
| | .*(?P<rationale>netflix\|itunes\|groove shark\|google music\|deezer\|spotify\|zvooq\|youtube\|lastfm\| pandora\|slacker\|iheart\|vimeo\|last fm).* |
| | .*(?P<rationale>animated movies\|films\|film).* |
| | .*(?P<rationale>twenties\|fourties\|eighties\|thirties\|sixties\|fifties\|seventies\|nineties\|1958\|2011\|2003\|2016) |
| | .*(?P<rationale>for\|at) (?P<rationale1>entertainment\|theatres\|corporation\|cinemas).* |
| | .*(?P<rationale>highly rated\|best\|popular\|top-rated\|top).* |
| | .*(?P<rationale>colder\|chilly\|warm\|hot\|freezing\|hotter\|cold\|warmer).* |
| | .*(?P<rationale>blizzard\|rain\|cloudy\|windy\|hail\|snowstorm\|stormy).* |
| | .*(?P<rationale>\bhere\|current position\|current location\|current place\|current spot).* |

Table 5: The upper Table refers to the dictionary we construct to match rationales for each intent type for Intent Classification task. The bottom one refers to Regular Expressions to match rationales for the Slot Filling task. Tokens following <rationale*> tag are annotated rationales.

| Setting | Method | seed | lr | lr_extractor | bz | max epochs | alpha | epsilon | warup epochs/ #multi-rounds | early stop | GPU |
|---|---|---|---|---|---|---|---|---|---|---|---|
| 1-shot | UIMER-Im +MASK | 55:1988:12333:42 | 0.0002 | - | 24 | 70 | 4 | 1 | - | 7 | |
| | UIMER-Im +BERT | 55:1988:12333:42 | 0.0004 | - | 24 | 70 | 4 | 0.5 | - | 7 | |
| | UIMER-Im +Prior | 55:1988:12333:42 | 0.0004 | - | 24 | 70 | 1 | 0.5 | - | 7 | |
| | UIMER-Im +Uniform | 55:1988:12333:42 | 0.0002 | - | 24 | 70 | 0.6 | 1 | - | 7 | |
| | UIMER-Im +MASK (warm.) | 55:1988:12333:42 | 0.0004 | - | 24 | 70 | 1 | 0.5 | 3 | 7 | Tesla V100-SXM2-32GB |
| | UIMER-Im +BERT (warm.) | 55:1988:12333:42 | 0.0002 | - | 24 | 70 | 4 | 0.05 | 2 | 7 | |
| | UIMER-Im +Prior (warm.) | 55:1988:12333:42 | 0.0004 | - | 24 | 70 | 4 | 0.5 | 2 | 7 | |
| | UIMER-Im +Uniform (warm.) | 55:1988:12333:42 | 0.0004 | - | 24 | 70 | 4 | 0.5 | 1 | 7 | |
| | UIMER-Dm One-pass | 55:1988:12333:42 | 0.0002 | 0.0005 | 24 | 70 | 0.8 | - | - | 7 | |
| | UIMER-Dm Multi-round | 55:1988:12333:42 | 0.0002 | 0.001 | 24 | 1 | 0.8 | - | 50 | 7 | |
| 3-shot | UIMER-Im +MASK | 55:1988:12333:42 | 0.0004 | - | 24 | 70 | 0.6 | 0.5 | - | 7 | |
| | UIMER-Im +BERT | 55:1988:12333:42 | 0.0004 | - | 24 | 70 | 0.6 | 0.5 | - | 7 | |
| | UIMER-Im +Prior | 55:1988:12333:42 | 0.0002 | - | 24 | 70 | 1 | 0.01 | - | 7 | |
| | UIMER-Im +Uniform | 55:1988:12333:42 | 0.0004 | - | 24 | 70 | 4 | 0.5 | - | 7 | |
| | UIMER-Im +MASK (warm.) | 55:1988:12333:42 | 0.0002 | - | 24 | 70 | 4 | 0.01 | 5 | 7 | Tesla V100-SXM2-32GB |
| | UIMER-Im +BERT (warm.) | 55:1988:12333:42 | 0.0004 | - | 24 | 70 | 1 | 0.05 | 5 | 7 | |
| | UIMER-Im +Prior (warm.) | 55:1988:12333:42 | 0.0002 | - | 24 | 70 | 1 | 1 | 5 | 7 | |
| | UIMER-Im +Uniform (warm.) | 55:1988:12333:42 | 0.0002 | - | 24 | 70 | 1 | 0.5 | 5 | 7 | |
| | UIMER-Dm One-pass | 55:1988:12333:42 | 0.0004 | 0.001 | 24 | 70 | 0.5 | - | - | 7 | |
| | UIMER-Dm Multi-round | 55:1988:12333:42 | 0.0002 | 0.0005 | 24 | 50 | 0.8 | - | 50 | 7 | |
| 10-shot | UIMER-Im +MASK | 55:1988:12333:42 | 0.0002 | - | 24 | 70 | 0.1 | 0.5 | - | 7 | |
| | UIMER-Im +BERT | 55:1988:12333:42 | 0.0001 | - | 24 | 70 | 0.1 | 0.1 | - | 7 | |
| | UIMER-Im +Prior | 55:1988:12333:42 | 0.0002 | - | 24 | 70 | 0.5 | 0.1 | - | 7 | |
| | UIMER-Im +Uniform | 55:1988:12333:42 | 0.0001 | - | 24 | 70 | 4 | 0.5 | - | 7 | |
| | UIMER-Im +MASK (warm.) | 55:1988:12333:42 | 0.0002 | - | 24 | 70 | 0.1 | 0.1 | 5 | 7 | Tesla V100-SXM2-32GB |
| | UIMER-Im +BERT (warm.) | 55:1988:12333:42 | 0.0002 | - | 24 | 70 | 4 | 0.01 | 5 | 7 | |
| | UIMER-Im +Prior (warm.) | 55:1988:12333:42 | 0.0002 | - | 24 | 70 | 4 | 0.01 | 10 | 7 | |
| | UIMER-Im +Uniform (warm.) | 55:1988:12333:42 | 0.0002 | - | 24 | 70 | 10 | 0.01 | task trained* | 7 | |
| | UIMER-Dm One-pass | 55:1988:12333:42 | 0.0004 | 0.001 | 24 | 70 | 0.5 | - | - | 7 | |
| | UIMER-Dm Multi-round | 55:1988:12333:42 | 0.0004 | 0.001 | 24 | 1 | 0.5 | - | 50 | 7 | |
| 30-shot | UIMER-Im +MASK | 55:1988:12333:42 | 0.0001 | - | 24 | 50 | 1 | 0.01 | - | 7 | |
| | UIMER-Im +BERT | 55:1988:12333:42 | 0.0001 | - | 24 | 50 | 0.6 | 0.01 | - | 7 | |
| | UIMER-Im +Prior | 55:1988:12333:42 | 0.0001 | - | 24 | 50 | 0.08 | 0.05 | - | 7 | |
| | UIMER-Im +Uniform | 55:1988:12333:42 | 0.0001 | - | 24 | 50 | 0.6 | 0.01 | - | 7 | |
| | UIMER-Im +MASK (warm.) | 55:1988:12333:42 | 0.0001 | - | 24 | 50 | 0.6 | 0.5 | 2 | 7 | NVIDIA TITAN V |
| | UIMER-Im +BERT (warm.) | 55:1988:12333:42 | 0.0001 | - | 24 | 50 | 0.08 | 0.05 | 1 | 7 | |
| | UIMER-Im +Prior (warm.) | 55:1988:12333:42 | 0.0001 | - | 24 | 50 | 1 | 0.01 | 4 | 7 | |
| | UIMER-Im +Uniform (warm.) | 55:1988:12333:42 | 0.0001 | - | 24 | 50 | 0.08 | 0.05 | 2 | 7 | |
| | UIMER-Dm One-pass | 55:1988:12333:42 | 0.0002 | 0.00001 | 24 | 25 | 0.1 | - | - | 7 | |
| | UIMER-Dm Multi-round | 55:1988:12333:42 | 0.0002 | 0.00001 | 24 | 25 | 0.1 | - | 15 | 7 | |
| full-resource | UIMER-Im +MASK | 55 | 0.00001 | - | 24 | 30 | 0.1 | 0.5 | - | 5 | |
| | UIMER-Im +BERT | 55 | 0.00001 | - | 24 | 30 | 0.01 | 0.05 | - | 5 | |
| | UIMER-Im +Prior | 55 | 0.00001 | - | 24 | 30 | 0.1 | 0.5 | - | 5 | |
| | UIMER-Im +Uniform | 55 | 0.00001 | - | 24 | 30 | 0.001 | 0.05 | - | 5 | |
| | UIMER-Im +MASK (warm.) | 55 | 0.00001 | - | 24 | 30 | 0.001 | 0.5 | 1 | 5 | NVIDIA TITAN V |
| | UIMER-Im +BERT (warm.) | 55 | 0.00001 | - | 24 | 30 | 0.1 | 0.5 | 5 | 5 | |
| | UIMER-Im +Prior (warm.) | 55 | 0.00001 | - | 24 | 30 | 0.001 | 0.5 | 1 | 5 | |
| | UIMER-Im +Uniform (warm.) | 55 | 0.00001 | - | 24 | 30 | 0.0001 | 0.05 | 5 | 5 | |
| | UIMER-Dm One-pass | 55 | 0.00001 | 0.001 | 24 | 30 | 0.1 | - | - | 5 | |
| | UIMER-Dm Multi-round | 55 | 0.00001 | 0.001 | 24 | 30 | 0.1 | - | 30 | 5 | |

Table 6: Hyperparameters for task Intent Classification. task trained*: The origin task is firstly well trained, then objective 1 is optimized.

| Setting | Method | seed | lr | lr_extractor | bz | max epochs | alpha | epsilon | warup epochs/ #multi-rounds | early stop | GPU |
|---|---|---|---|---|---|---|---|---|---|---|---|
| 1-shot | UIMER-IM +MASK | 55:1988:12333:42 | 0.0002 | - | 24 | 70 | 0.2 | 0.5 | - | 7 | |
| | UIMER-IM +BERT | 55:1988:12333:42 | 0.0002 | - | 24 | 70 | 1 | 1 | - | 7 | |
| | UIMER-IM +Prior | 55:1988:12333:42 | 0.0002 | - | 24 | 70 | 0.6 | 1 | - | 7 | |
| | UIMER-IM +Uniform | 55:1988:12333:42 | 0.0002 | - | 24 | 70 | 0.6 | 0.5 | - | 7 | |
| | UIMER-IM +MASK (warm.) | 55:1988:12333:42 | 0.0002 | - | 24 | 70 | 0.08 | 0.5 | 5 | 7 | |
| | UIMER-IM +BERT (warm.) | 55:1988:12333:42 | 0.0002 | - | 24 | 70 | 0.6 | 0.01 | 5 | 7 | NVIDIA TITAN V |
| | UIMER-IM +Prior (warm.) | 55:1988:12333:42 | 0.0002 | - | 24 | 70 | 1 | 0.5 | 5 | 7 | |
| | UIMER-IM +Uniform (warm.) | 55:1988:12333:42 | 0.0002 | - | 24 | 70 | 1 | 0.05 | 5 | 7 | |
| | UIMER-DM One-pass | 55:1988:12333:42 | 0.0002 | 0.001 | 24 | 70 | 0.5 | - | - | 7 | |
| | UIMER-DM Multi-round | 55:1988:12333:42 | 0.0002 | 0.001 | 24 | 70 | 0.5 | - | 10 | 7 | |
| 3-shot | UIMER-IM +MASK | 55:1988:12333:42 | 0.0001 | - | 24 | 70 | 1 | 4 | - | 7 | |
| | UIMER-IM +BERT | 55:1988:12333:42 | 0.0001 | - | 24 | 70 | 0.08 | 0.1 | - | 7 | |
| | UIMER-IM +Prior | 55:1988:12333:42 | 0.0001 | - | 24 | 70 | 0.1 | 2 | - | 7 | |
| | UIMER-IM +Uniform | 55:1988:12333:42 | 0.0001 | - | 24 | 70 | 1 | 4 | - | 7 | |
| | UIMER-IM +MASK (warm.) | 55:1988:12333:42 | 0.0001 | - | 24 | 70 | 0.005 | 2 | 3 | 7 | |
| | UIMER-IM +BERT (warm.) | 55:1988:12333:42 | 0.0001 | - | 24 | 70 | 1 | 2 | task trained* | 7 | NVIDIA TITAN V |
| | UIMER-IM +Prior (warm.) | 55:1988:12333:42 | 0.0001 | - | 24 | 70 | 0.08 | 4 | 1 | 7 | |
| | UIMER-IM +Uniform (warm.) | 55:1988:12333:42 | 0.0001 | - | 24 | 70 | 1 | 4 | 1 | 7 | |
| | UIMER-DM One-pass | 55:1988:12333:42 | 0.0001 | 0.003 | 24 | 70 | 0.1 | - | - | 7 | |
| | UIMER-DM Multi-round | 55:1988:12333:42 | 0.0001 | 0.003 | 24 | 70 | 2 | - | 3 | 3 | |
| 10-shot | UIMER-IM +MASK | 55:1988:12333:42 | 0.001 | - | 24 | 70 | 4 | 4 | - | 7 | |
| | UIMER-IM +BERT | 55:1988:12333:42 | 0.001 | - | 24 | 70 | 0.01 | 2 | - | 7 | |
| | UIMER-IM +Prior | 55:1988:12333:42 | 0.001 | - | 24 | 70 | 0.6 | 4 | - | 7 | |
| | UIMER-IM +Uniform | 55:1988:12333:42 | 0.001 | - | 24 | 70 | 4 | 4 | - | 7 | |
| | UIMER-IM +MASK (warm.) | 55:1988:12333:42 | 0.001 | - | 24 | 70 | 4 | 4 | 2 | 7 | |
| | UIMER-IM +BERT (warm.) | 55:1988:12333:42 | 0.001 | - | 24 | 70 | 0.01 | 4 | 1 | 7 | NVIDIA TITAN V |
| | UIMER-IM +Prior (warm.) | 55:1988:12333:42 | 0.001 | - | 24 | 70 | 4 | 2 | 2 | 7 | |
| | UIMER-IM +Uniform (warm.) | 55:1988:12333:42 | 0.001 | - | 24 | 70 | 6 | 2 | 1 | 7 | |
| | UIMER-DM One-pass | 55:1988:12333:42 | 0.00001 | 0.0001 | 24 | 70 | 1 | - | - | 7 | |
| | UIMER-DM Multi-round | 55:1988:12333:42 | 0.00001 | 0.0001 | 24 | 10 | 1 | - | 10 | 7 | |
| 30-shot | UIMER-IM +MASK | 55:1988:12333:42 | 0.0001 | - | 24 | 70 | 0.5 | 0.01 | - | 10 | |
| | UIMER-IM +BERT | 55:1988:12333:42 | 0.0001 | - | 24 | 50 | 1 | 0.5 | - | 10 | |
| | UIMER-IM +Prior | 55:1988:12333:42 | 0.0001 | - | 24 | 50 | 0.6 | 0.01 | - | 10 | |
| | UIMER-IM +Uniform | 55:1988:12333:42 | 0.0001 | - | 24 | 50 | 1 | 0.5 | - | 10 | |
| | UIMER-IM +MASK (warm.) | 55:1988:12333:42 | 0.0001 | - | 24 | 50 | 10 | 0.05 | 1 | 10 | |
| | UIMER-IM +BERT (warm.) | 55:1988:12333:42 | 0.00008 | - | 24 | 50 | 0.001 | 0.05 | 4 | 10 | NVIDIA TITAN V |
| | UIMER-IM +Prior (warm.) | 55:1988:12333:42 | 0.00008 | - | 24 | 50 | 0.001 | 0.05 | 2 | 10 | |
| | UIMER-IM +Uniform (warm.) | 55:1988:12333:42 | 0.00008 | - | 24 | 50 | 1 | 0.05 | 1 | 10 | |
| | UIMER-DM One-pass | 55:1988:12333:42 | 0.0001 | 0.0001 | 24 | 20 | 0.08 | - | - | 10 | |
| | UIMER-DM Multi-round | 55:1988:12333:42 | 0.0001 | 0.0001 | 24 | 10 | 0.08 | - | 20 | 10 | |
| full-resource | UIMER-IM +MASK | 55 | 0.00001 | - | 24 | 20 | 0.1 | 1 | - | 5 | |
| | UIMER-IM +BERT | 55 | 0.00001 | - | 24 | 20 | 0.01 | 1 | - | 5 | |
| | UIMER-IM +Prior | 55 | 0.00001 | - | 24 | 20 | 0.1 | 1 | - | 5 | |
| | UIMER-IM +Uniform | 55 | 0.00001 | - | 24 | 20 | 0.0001 | 1 | - | 5 | |
| | UIMER-IM +MASK (warm.) | 55 | 0.00001 | - | 24 | 20 | 0.1 | 1 | 3 | 5 | |
| | UIMER-IM +BERT (warm.) | 55 | 0.00001 | - | 24 | 20 | 0.0001 | 1 | 1 | 5 | NVIDIA TITAN V |
| | UIMER-IM +Prior (warm.) | 55 | 0.00001 | - | 24 | 20 | 0.1 | 1 | 1 | 5 | |
| | UIMER-IM +Uniform (warm.) | 55 | 0.00001 | - | 24 | 20 | 0.0001 | 1 | 3 | 5 | |
| | UIMER-DM One-pass | 55 | 0.00001 | 0.001 | 24 | 20 | 20 | - | - | 5 | |
| | UIMER-DM Multi-round | 55 | 0.00001 | 0.001 | 24 | 20 | 20 | - | 50 | 5 | |

Table 7: Hyperparameters for task Slot Filling. task trained*: The origin task is firstly well trained, then objective 1 is optimized.

| Setting | Method | seed | lr | lr_extractor | bz | max epochs | alpha | epsilon | warup epochs/ #multi-rounds | early stop | GPU |
|---|---|---|---|---|---|---|---|---|---|---|---|
| 100 | UIMER-IM +MASK | 55:1988:12333:42 | 0.0003 | - | 32 | 50 | 1 | 1 | - | 8 | |
| | UIMER-IM +BERT | 55:1988:12333:42 | 0.0003 | - | 32 | 50 | 0.1 | 0.01 | - | 8 | |
| | UIMER-IM +Prior | 55:1988:12333:42 | 0.0003 | - | 32 | 50 | 0.001 | 0.01 | - | 8 | |
| | UIMER-IM +Uniform | 55:1988:12333:42 | 0.0003 | - | 32 | 50 | 0.1 | 0.1 | - | 8 | |
| | UIMER-IM +MASK (warm.) | 55:1988:12333:42 | 0.0003 | - | 32 | 50 | 1 | 0.01 | 0.1* | 8 | |
| | UIMER-IM +BERT (warm.) | 55:1988:12333:42 | 0.0003 | - | 32 | 50 | 0.01 | 0.1 | 0.1 | 8 | NVIDIA A40 |
| | UIMER-IM +Prior (warm.) | 55:1988:12333:42 | 0.0003 | - | 32 | 50 | 0.1 | 0.01 | 0.1 | 8 | |
| | UIMER-IM +Uniform (warm.) | 55:1988:12333:42 | 0.0003 | - | 32 | 50 | 0.001 | 0.01 | 0.1 | 8 | |
| | UIMER-DM One-pass | 55:1988:12333:42 | 0.0003 | 0.001 | 24 | 30 | 0.01 | - | - | 8 | |
| | UIMER-DM Multi-round | 55:1988:12333:42 | 0.0003 | 0.001 | 24 | 30 | 0.01 | - | 10 | 8 | |
| 500 | UIMER-IM +MASK | 55:1988:12333:42 | 0.001 | - | 16 | 50 | 10 | 0.01 | - | 5 | |
| | UIMER-IM +BERT | 55:1988:12333:42 | 0.00008 | - | 16 | 50 | 1 | 0.01 | - | 5 | |
| | UIMER-IM +Prior | 55:1988:12333:42 | 0.0001 | - | 8 | 50 | 0.01 | 0.1 | - | 5 | |
| | UIMER-IM +Uniform | 55:1988:12333:42 | 0.0001 | - | 16 | 50 | 1 | 1 | - | 5 | |
| | UIMER-IM +MASK (warm.) | 55:1988:12333:42 | 0.00008 | - | 8 | 50 | 10 | 0.01 | 1 | 30 | |
| | UIMER-IM +BERT (warm.) | 55:1988:12333:42 | 0.0001 | - | 16 | 50 | 0.1 | 1 | 1 | 30 | NVIDIA TITAN Xp |
| | UIMER-IM +Prior (warm.) | 55:1988:12333:42 | 0.00008 | - | 8 | 50 | 1 | 0.01 | 1 | 30 | |
| | UIMER-IM +Uniform (warm.) | 55:1988:12333:42 | 0.00008 | - | 8 | 50 | 1 | 1 | 1 | 30 | |
| | UIMER-DM One-pass | 55:1988:12333:42 | 0.00008 | 0.001 | 16 | 30 | 0.01 | - | - | 8 | |
| | UIMER-DM Multi-round | 55:1988:12333:42 | 0.00008 | 0.001 | 16 | 30 | 0.01 | - | 3 | 8 | |

Table 8: Hyperparameters for task NLI. 0.1*: 10% of the whole mini-batches are used to do warm-up training.

| | | IC | | | | | | | | SF | | | | | | | | NLI | | | |
| | | 1 | | 3 | | 10 | | 30 | | 1 | | 3 | | 10 | | 30 | | 100 | | 500 | |
| | Model | mean | std | mean | std | mean | std | mean | std | mean | std | mean | std | mean | std | mean | std | mean | std | mean | std |
|---|---|---|---|---|---|---|---|---|---|---|---|---|---|---|---|---|---|---|---|---|---|
| | Baseline | 65.71 | 7.30 | 79.18 | 6.17 | 91.00 | 1.40 | 93.79 | 0.62 | 38.14 | 1.92 | 50.97 | 1.23 | 67.05 | 0.58 | 81.70 | 0.12 | 54.03 | 13.66 | 62.84 | 7.38 |
| | UIMER-G$_B$ Ghaeini et al. (2019) | 65.71 | 7.30 | 79.14 | 6.12 | 91.82 | 0.18 | 94.18 | 1.30 | 37.77 | 1.22 | 51.69 | 0.77 | 67.57 | 0.47 | 82.16 | 0.57 | 67.13 | 0.86 | 69.77 | 2.51 |
| Our Framework | UIMER-G$_B$ Huang et al. (2021) + base | 67.04 | 9.11 | 83.04 | 4.83 | 91.43 | 1.14 | 94.57 | 0.20 | 39.02 | 1.57 | 50.66 | 0.98 | 67.11 | 1.22 | 80.20 | 2.64 | 66.15 | 1.37 | 68.57 | 2.76 |
| | + gate | 67.14 | 5.19 | 82.01 | 3.19 | 91.39 | 0.05 | 94.07 | 1.02 | 37.84 | 1.77 | 51.63 | 1.76 | 67.34 | 0.53 | 80.89 | 2.24 | 66.06 | 1.13 | 68.31 | 4.22 |
| | + order | 65.28 | 4.71 | 81.82 | 4.35 | 90.82 | 1.21 | 94.36 | 1.38 | 38.18 | 1.64 | 50.99 | 1.46 | 67.55 | 0.68 | 81.08 | 1.25 | 68.44 | 3.50 | 68.57 | 2.76 |
| | + (gate+order) | 67.71 | 6.24 | 80.25 | 5.57 | 92.11 | 1.46 | 94.39 | 1.07 | 38.86 | 1.83 | 51.73 | 1.89 | 67.76 | 0.76 | 80.55 | 1.30 | 65.10 | 7.62 | 65.84 | 4.22 |
| | UIMER-Im + MASK | 69.85 | 4.24 | 83.17 | 5.32 | 91.18 | 0.50 | 93.86 | 1.32 | 38.68 | 2.73 | 52.47 | 1.52 | 69.27 | 1.21 | 81.67 | 1.90 | 63.36 | 1.19 | 70.04 | 5.22 |
| | + BERT | 70.61 | 2.52 | 83.93 | 3.69 | 91.78 | 1.16 | 94.61 | 0.64 | 39.28 | 1.80 | 51.96 | 1.18 | 69.22 | 1.92 | 81.85 | 1.02 | 62.08 | 6.04 | 69.03 | 4.70 |
| | + Prior | 73.71 | 4.75 | 83.93 | 5.68 | 91.00 | 0.63 | 93.96 | 0.43 | 38.07 | 1.21 | 51.69 | 0.88 | 68.31 | 0.71 | 81.97 | 1.87 | 66.56 | 2.58 | 70.32 | 2.99 |
| | + Uniform | 73.32 | 4.02 | 86.04 | 2.16 | 91.64 | 1.48 | 94.00 | 0.60 | 39.31 | 2.14 | 51.37 | 1.82 | 69.02 | 0.89 | 81.69 | 1.91 | 64.34 | 7.87 | 69.19 | 2.20 |
| | UIMER-Im + MASK (warm.) | 70.82 | 2.40 | 82.96 | 5.11 | 92.43 | 1.59 | 94.04 | 0.54 | 38.60 | 2.23 | 51.55 | 1.35 | 67.93 | 0.90 | 82.25 | 1.38 | 68.79 | 2.44 | 69.95 | 0.78 |
| | + BERT (warm.) | 70.93 | 5.56 | 82.71 | 4.07 | 92.00 | 1.64 | 94.32 | 0.11 | 39.53 | 1.73 | 52.83 | 1.18 | 68.81 | 1.85 | 82.58 | 0.96 | 68.89 | 0.88 | 69.18 | 1.51 |
| | + Prior (warm.) | 73.82 | 5.97 | 83.11 | 2.19 | 91.93 | 1.08 | 94.07 | 0.74 | 38.24 | 2.27 | 51.68 | 1.03 | 68.26 | 1.38 | 82.66 | 1.32 | 68.25 | 3.35 | 71.82 | 3.57 |
| | + Uniform (warm) | 75.79 | 4.94 | 86.29 | 1.51 | 91.67 | 1.21 | 94.32 | 0.80 | 38.71 | 2.11 | 52.18 | 0.71 | 68.21 | 0.46 | 82.17 | 1.54 | 67.58 | 1.77 | 69.79 | 1.30 |
| | UIMER-Dm One-pass | 66.75 | 7.54 | 84.42 | 5.42 | 91.53 | 0.58 | 93.78 | 0.62 | 39.86 | 1.53 | 52.87 | 2.25 | 67.28 | 0.71 | 81.90 | 1.40 | 63.03 | 4.92 | 66.91 | 4.77 |
| | Multi-round | 70.21 | 8.57 | 85.86 | 4.40 | 91.92 | 0.92 | 94.00 | 0.60 | 41.32 | 1.17 | 53.10 | 0.81 | 69.26 | 0.53 | 82.00 | 1.54 | 65.44 | 2.94 | 67.60 | 8.59 |

Table 9: Result with std. on all few-shot settings.