# OpenReview forum: "Using Interpretation Methods for Model Enhancement"
_EMNLP/2023/Conference — EMNLP 2023 Main_

### Official Review · Reviewer_PcW8 · 2023-08-04

**Soundness:** 4

**Excitement:**

3: Ambivalent: It has merits (e.g., it reports state-of-the-art results, the idea is nice), but there are key weaknesses (e.g., it describes incremental work), and it can significantly benefit from another round of revision. However, I won't object to accepting it if my co-reviewers champion it.

**Paper Topic And Main Contributions:**

In situations where gold rationales can be obtained for a task, the authors demonstrate how to use these rationales as an additional source of supervision while training the models. Model interpretation techniques allow us to gauge the level of "importance" the model places on each token. These feature importances are used to gauge the model's rationale. When training the model, the authors add a new task to the objective function to measure the distance between the model's rationale (obtained by model interpretation techniques) and the gold rationale (either 0 or 1 for each token).

Their main contributions are -
1. They demonstrate how to implement their idea of using gold rationales to provide additional supervision by using three different existing model interpretation techniques. They try different (minor) variations when using each model interpretation technique, providing the reader with some guidelines for how to squeeze out the best performance.
2. Using three different datasets, the authors demonstrate that their technique outperforms the baseline (not using gold rationales) as well as existing techniques that use similar training strategies. Furthermore, they provide empirical evidence showing that the gain in performance is larger in low-resource settings.

**Reasons To Accept:**

1. The paper demonstrates how to use model interpretation techniques and gold rationales to improve the performance of the model on certain tasks.
2. Their method produces results (on three datasets) that are better than existing techniques that try to use gold rationales.
3. I appreciate that they talk about how they obtained the gold rationales since these aren't often present in datasets and can be a major practical limitation. It is promising to see that even simple techniques to create gold rationales led to good improvement in the results.

**Reasons To Reject:**

While I appreciate that they demonstrate how to use three different model interpretation techniques in their framework, calling the idea a "framework" seems a bit misleading. This is mostly because frameworks often have multiple components in the system that are novel yet remain constant across different implementations. Whereas, in this paper, the main part of the framework, the "L_int" function changes for each model interpretation technique.

Section 5.3 seems to be based on just a few examples and doesn't seem to justify the authors' claims that their method is able to enhance the model in performance as well as interpretability.

**Reproducibility:**

4: Could mostly reproduce the results, but there may be some variation because of sample variance or minor variations in their interpretation of the protocol or method.

**Reviewer Confidence:**

4: Quite sure. I tried to check the important points carefully. It's unlikely, though conceivable, that I missed something that should affect my ratings.

**Typos Grammar Style And Presentation Improvements:**

Please consider increasing the label font size for the x and y axis labels in Figure 3.

---

> ### Author Rebuttal · Authors · 2023-08-29
>
> Thanks for your suggestions for our work. As for your concern:
>
> Framework:
>
> - We would like to argue that this is just a matter of wording and should not be a reason for rejection. We choose the word “framework” because different interpretation methods can be plugged into our “framework” following the same setup as defined at the beginning of Sec. 3. We could perhaps replace “framework” with “paradigm”.
>
> Sec. 5.3:
>
> Sec. 5.3 is just a **case study** and by nature it contains only a few representative examples. As for more rigorous support for our claims:
>
> 1. Sec. 4.2 “Main Result” shows that our framework is able to enhance models in performance.
> 2. Sec. 5.4 shows that our framework is also able to enhance interpretability (in particular, the third column (% $a_R > a_N$) of Table 4). We can see that with UIMER-Im and -Dm, rationales in the test sets are better recognized (i.e., more rationale tokens are given higher attribution scores).

---

### Official Review · Reviewer_fRry · 2023-08-05

**Soundness:** 3

**Excitement:**

2: Mediocre: This paper makes marginal contributions (vs non-contemporaneous work), so I would rather not see it in the conference.

**Missing References:**

[1] Zhang et al., Rationale-augmented convolutional neural networks for text classification, 2016.

[2] Liu and Avci, Incorporating priors with feature attribution on text classification, 2019.

[3] Rieger et al., Interpretations are useful: penalizing explanations to align neural networks with prior knowledge, 2019.

[4] Sekhon et al., Improving Interpretability via Explicit Word Interaction Graph Layer, 2023.

[5] Du et al., Learning credible deep neural networks with rationale regularization, 2019.


**Paper Topic And Main Contributions:**

This paper proposes a framework called UIMER that utilizes interpretation methods and gold rationales to improve model performance. Interpretation methods (e.g., gradient-based methods, erasure/replace-based methods, and extractor-based methods) identify important features in the input, which are then aligned with gold rationales during training. The overall objective combines the original task-specific learning objective and the interpretation alignment objective. A BERT-base model is adopted as the baseline model and tested with UIMER using different interpretation methods on three tasks (Intent Classification, Slot Filling, and Natural Language Inference). Various training paradigms are investigated across few-shot and full-data settings.

**Reasons To Accept:**

-	The idea of utilizing interpretations to improve model performance is promising.
-	The paper is well-written and easy to follow.
-	A comprehensive study (with different interpretation methods and training paradigms) is conducted.


**Reasons To Reject:**

-	While this work contributes valuable insights, its novelty appears limited. The utilization of interpretations to enhance model performance has been extensively explored in both supervised and unsupervised approaches [1-7].
-	The proposed method, which relies on gold rationales as additional supervision, may pose limitations in real-world applications due to the expense of collecting high-quality rationales.
-	Only the BERT-base model is tested. It would be interesting to see how the proposed method generalizes to other model architectures.
-	The performance improvement, particularly when using full data, appears to be marginal. Further investigation into the underlying reasons for this observation would be valuable.

[1] Zhang et al., Rationale-augmented convolutional neural networks for text classification, 2016.

[2] Liu and Avci, Incorporating priors with feature attribution on text classification, 2019.

[3] Ross et al., Right for the right reasons: Training differentiable models by constraining their explanations, 2017.

[4] Rieger et al., Interpretations are useful: penalizing explanations to align neural networks with prior knowledge, 2019.

[5] Du et al., Learning credible deep neural networks with rationale regularization, 2019.

[6] Chen and Ji., Learning Variational Word Masks to Improve the Interpretability of Neural Text Classifiers, 2020.

[7] Sekhon et al., Improving Interpretability via Explicit Word Interaction Graph Layer, 2023.



**Reproducibility:**

3: Could reproduce the results with some difficulty. The settings of parameters are underspecified or subjectively determined; the training/evaluation data are not widely available.

**Reviewer Confidence:**

4: Quite sure. I tried to check the important points carefully. It's unlikely, though conceivable, that I missed something that should affect my ratings.

---

> ### Author Rebuttal · Authors · 2023-08-29
>
> Thanks for your detailed reply. As for your concern:
>
> Reasons To Reject:
>
> - [1-7]:
>
>     We want to clarify that our work is valuable and novel despite [1-7].
>
>     [1,4] do not utilize interpretation methods and solely utilize rationales.
>
>     [6,7] focus on improving model interpretability and do not leverage gold rationales for model enhancement, which does not match our motivation or setup.
>
>     [2,3,5] are all instances of our framework. [2,3] belong to UIMER-Gb and are similar to previous work discussed in Sec. 3.1. As we mentioned in the paper, UIMER-Gb is not our contribution.
>
>     [5] is similar to UIMER-Im+MASK and is the only work among [1-7] that is closely related to one of our main contributions (i.e., UIMER-Im). However, [5] has the OOD problem and does not outperform baselines in their experiments (Table III of [5]). In contrast, our UIMER-Im under various settings consistently outperforms the baseline in our experiments.
>
>     Most of the works only experiment on classification tasks [1,2,5,6,7] in rich-resource settings [1-7] and we comprehensively evaluate classification, slot filling and natural language inference tasks in both low and rich-resource settings.
>
>
>     Below we discuss each reference in more detail.
>
>     - [1] is a work that utilizes information stored in rationales but not a work that utilizes interpretation methods. An extra module apart from the task model is responsible for extracting rationales and it is trained to align gold rationales. However: 1) there are no concepts of utilizing interpretation methods because the extra module just extracts tokens as how it is trained by rot memorization.
>
>     - [2] and [3] can also be seen as instances as we discussed in Sec. 2.2, similar to Ghaeini et al. (2019) and Huang et al. (2021). In [2] the authors apply “Integrated Gradients” techniques that are also discussed in our paper (line 125).
>
>     - [4] uses the “Context Decomposition (CD)” method to decompose logits $g(x)$ into two terms and view one of them ($\beta$) “importance score”. CD was first introduced by “Murdoch et al., Beyond Word Importance: Contextual Decomposition To Extract Interactions From LSTMs”. Rieger et al. modify the method to apply to other model architectures (CNNs).
>
>         1) Obviously, it is model-dependent while our framework is model-agnostic.
>
>         2) The decomposition process still remains black-box: $\beta$ is still not interpretable when it is defined. Though it can memorize some rationales after supervised training, it is not able to find the tokens that influence the model’s output the most. In our work, all interpretation methods are interpretable themselves without the need for supervision from gold rationales/explanations.
>
>         3) We further propose “Multi-Round” training strategy (Sec. 5.2) that better coordinates task loss and interpretation loss (refer to the definition in Equation 1 in [4]).
>
>     - [5] calculates the importance score $s^t$ by measuring the deviation of the prediction between the original input and the partial input $x$ with token $t$ omitted which suffers **Out-Of-Distribution (OOD) problem** discussed by “Kim et al., Interpretation of NLP models through input marginalization (the third paragraph in their Introduction)”. From Table III in [5], the method does not outperform the baseline methods in half of the settings.
>
>     - [6] and [7] focus on improving model interpretability and do not leverage gold rationales for model enhancement. Compared to [6], we further propose “Multi-Round” training strategy that better aligns the “interpretation module” and “task model”, which results in better performance than “one-pass” training (Sec. 5.2). As for [7], they design a new layer that can discover interactions between words. However, we do not aim to develop new interpretation methods/modules, instead, we aim to utilize interpretation methods and gold rationales for model enhancement.
>
>
> - High-quality rationales:
>
>     The approaches we use to obtain rationales are simple and fast as stated in “Sec. 4.1, Rationales”. Despite the rationales being not perfect, we still find that just simple rationales and proper utilization of various interpretation methods are able to enhance models.
>
> - Other model architectures:
>
>     It is interesting to use UIMER on other base models, which we leave as future work.
>
> - Performance:
>
>     First of all, we would like to point out that our performance improvement over the baseline is quite consistent in low-resource settings (i.e., not using full data), as shown in Table 2 and also Table 9 (showing standard deviations). Second, a much smaller improvement on full data is to be expected. The table below shows that with full training data on Intent Classification, the base model can already achieve near-perfect task accuracy as well as very high interpretation accuracy (90.34%), so our framework has little room to further improve the task accuracy by increasing the interpretation accuracy.
>
>     |  |  | Acc. (a_R > a_N) | Acc. (a_R < a_N) | % |
>     | --- | --- |:---: |:---: |:---: |
>     | BaseModel +Im | 1shot | 68.86 | 37.86 | 81.57 |
>     | BaseModel +Im | full | 99.01 | 98.15 | 90.34 |
>     | UIMER +Im | 1shot | 77.99 | 36.59 | 85.33 |
>     | UIMER +Im | full | 99.46 | 33.33 | 99.46 |
>
>     Note that “Acc. ($a_R$ * $a_N$)” shown here is slightly different from Table 2 because the Acc. on samples without gold rationale cannot be included ($a_R$ and $a_N$ cannot be calculated without gold rationales annotated in the sample.)

---

### Official Review · Reviewer_xBwV · 2023-08-13

**Soundness:** 4

**Excitement:**

3: Ambivalent: It has merits (e.g., it reports state-of-the-art results, the idea is nice), but there are key weaknesses (e.g., it describes incremental work), and it can significantly benefit from another round of revision. However, I won't object to accepting it if my co-reviewers champion it.

**Paper Topic And Main Contributions:**

This paper proposes a general framework for utilizing model interpretation to enhance model performance. The main contribution of this paper lies in the exploration of utilizing two novel interpretation methods to enhance model performance and extensive evaluation on various tasks is conducted. Compared with existing papers in this area, this study designs a more general framework and conducts experiments in more tasks and datasets.

**Questions For The Authors:**

[Q1] Baselines: Are there other stronger baselines for the tasks like intent classification? The practical impact of this paper might be decreased if a stronger baseline outperforms the combination of the proposed method + a weak base model. Moreover, it is still possible that the proposed method cannot bring an improvement to the stronger baseline model.

[Q2] Experimental metrics. The metrics seem to include standard metrics like F1. However, the reviewer is wondering whether the generated interpretation is also measured and serves as another metric as shown in Figure 1 in Introduction?


**Reasons To Accept:**

[+] Research topic. Model interpretation is an emerging topic. Also, utilizing interpretation to enhance model performance is an interesting direction.

[+] Experiments and evaluation. Extensive experiments on three tasks with two datasets are conducted. Detailed analysis is provided with a concrete case study to demonstrate the effectiveness of the proposed framework.

[+] Presentation. The paper is well-presented with proper figures and tables. The introduction is easy to follow to get the key idea in a short time.


**Reasons To Reject:**

[-] Experiment: The experiments focus on low-resource setting. However, it would be interesting to see some results when there are large-scale of training data. Even though rationales might not help improve the overall performance in settings with lots of training data, it would be interesting to explore some other problems, e.g., would the performance on out-of-distribution samples can be improved with external resources?

[-] Model: The challenges solved by the proposed model seems not introduced accordingly. Even though the authors argue that SoA methods do not compare different interpretation methods on different tasks, the missing of proper justification of challenges would decrease the technical contribution and novelty of this paper.

[-] Presentation: In Table 1, it would be better to use full names (e.g., intent classification) of tasks because there is enough space to make it easier to read with full names.


**Reproducibility:**

3: Could reproduce the results with some difficulty. The settings of parameters are underspecified or subjectively determined; the training/evaluation data are not widely available.

**Reviewer Confidence:**

3: Pretty sure, but there's a chance I missed something. Although I have a good feel for this area in general, I did not carefully check the paper's details, e.g., the math, experimental design, or novelty.

---

> ### Author Rebuttal · Authors · 2023-08-29
>
> Thanks for your suggestions. As for your concern:
>
> [-] Experiment:
>
> - We indeed evaluated UIMER in rich-resource settings (i.e., full data of the IC and SF tasks, results shown in the two “full” columns in Table 2) and almost all methods in UIMER outperform the base model. Experiments on even larger-scale training data and out-of-distribution settings are interesting future work.
>
> [-] Model:
>
> - This work is not trying to solve specific “challenges”. Instead, we aim to 1) generalize previous work into a framework that can utilize various interpretation methods to enhance models, 2) propose two novel instances of the framework, and 3) perform a more comprehensive empirical evaluation.
>
> Q1:
>
> - No matter how strong a baseline is, there would always be tasks and (low-resource) settings in which additional knowledge embodied in rationales is helpful, and that’s when our framework can be useful. Our experiments showcase such settings using the most widely used baselines.
>
> Q2:
>
> - The third column (% $a_R > a_N$) of Table 4 in Sec. 5.4 shows an evaluation of generated interpretations. We can see that with UIMER-Im and -Dm, rationales in the test sets are better recognized (i.e., more rationale tokens are given higher attribution scores). For your reference, we further measured the same metric of UIMER-Im in the rich-resource setting on Intent Classification task and found a consistent conclusion:
>     |  |  | Acc. ($a_R > a_N$) | Acc. ($a_R < a_N$) | % |
>     | --- | --- | :---: | :---: | :---: |
>     | BaseModel +Im | 1shot | 68.86 | 37.86 | 81.57 |
>     | BaseModel +Im | full | 99.01 | 98.15 | 90.34 |
>     | UIMER +Im | 1shot | 77.99 | 36.59 | 85.33 |
>     | UIMER +Im | full | 99.46 | 33.33 | 99.46 |
>
>     Note that “Acc. ($a_R$ * $a_N$)” shown here is slightly different from Table 2 because the Acc. on samples without gold rationale cannot be included ($a_R$ and $a_N$ cannot be calculated without gold rationales annotated in the sample.)

---

### Meta-Review · Area_Chair_8naq · 2023-09-15

**Recommendation:** 4

**Metareview:**

This paper presents a general framework, UIMER, which leverages model interpretation techniques and gold rationales to enhance model performance. The study explores various interpretation methods (e.g., gradient-based, erasure/replace-based, extractor-based) to identify key input features, aligning them with gold rationales during training. The framework combines task-specific objectives with interpretation alignment objectives. Experimental evaluation spans multiple tasks and datasets, showcasing its superiority over baseline models and existing techniques, particularly in low-resource scenarios.

The paper tackles the promising area of using model interpretation to boost model performance, showing potential in this domain. It excels in conducting extensive experiments across diverse tasks and datasets, making it well-structured and informative.
Nonetheless, it predominantly focuses on low-resource settings, leaving questions about its scalability to larger training data and adaptability to out-of-distribution scenarios. Addressing these aspects could enhance the paper's relevance and impact.

---

### Decision · Program_Chairs · 2023-10-07

**Decision:**

Accept-Main

**Comment:**

This paper presents a general framework, UIMER, which leverages model interpretation techniques and gold rationales to enhance model performance. The study explores various interpretation methods (e.g., gradient-based, erasure/replace-based, extractor-based) to identify key input features, aligning them with gold rationales during training. The framework combines task-specific objectives with interpretation alignment objectives. Experimental evaluation spans multiple tasks and datasets, showcasing its superiority over baseline models and existing techniques, particularly in low-resource scenarios.

The paper tackles the promising area of using model interpretation to boost model performance, showing potential in this domain. It excels in conducting extensive experiments across diverse tasks and datasets, making it well-structured and informative.
Nonetheless, it predominantly focuses on low-resource settings, leaving questions about its scalability to larger training data and adaptability to out-of-distribution scenarios. Addressing these aspects could enhance the paper's relevance and impact.